# From Extinction Learning to Anxiety Treatment: Mind the Gap

**DOI:** 10.3390/brainsci9070164

**Published:** 2019-07-11

**Authors:** Joseph K. Carpenter, Megan Pinaire, Stefan G. Hofmann

**Affiliations:** Department of Psychological and Brain Sciences, Boston University, 900 Commonwealth Ave, 2nd floor, Boston, MA 02215, USA

**Keywords:** extinction, fear, anxiety, conditioning, translational research, exposure therapy

## Abstract

Laboratory models of extinction learning in animals and humans have the potential to illuminate methods for improving clinical treatment of fear-based clinical disorders. However, such translational research often neglects important differences between threat responses in animals and fear learning in humans, particularly as it relates to the treatment of clinical disorders. Specifically, the conscious experience of fear and anxiety, along with the capacity to deliberately engage top-down cognitive processes to modulate that experience, involves distinct brain circuitry and is measured and manipulated using different methods than typically used in laboratory research. This paper will identify how translational research that investigates methods of enhancing extinction learning can more effectively model such elements of human fear learning, and how doing so will enhance the relevance of this research to the treatment of fear-based psychological disorders.

## 1. Introduction

Fear-based disorders are among the most prevalent categories of psychiatric conditions [1]. Although effective treatments exist for these disorders, response rates remain suboptimal [2,3]. One promising direction for improving such treatments is to translate insights from basic research on fear learning and memory into modified or novel clinical approaches. In particular, laboratory research on fear conditioning and extinction learning can serve as a useful translational model for understanding the factors underlying the development and elimination of maladaptive fear responses. In turn, such insights can illuminate potential methods of improving treatments of fear-based disorders [4]. 

Fear conditioning and extinction learning hold particular promise for translational research for a number of reasons. For one, the procedures involved in laboratory fear extinction studies approximate those used clinically in exposure therapy, which is one of the most effective treatments of anxiety disorders [2,3]. In extinction training, a conditioned stimulus (CS) that was previously paired with an aversive unconditioned stimulus (US) such as an electric shock is repeatedly presented without the US, resulting in a decrement in fear responses over time. Similarly, in exposure therapy, repeatedly approaching a feared situation, stimulus, or memory (i.e., a CS) allows patients to learn that their feared consequences (the US) are unlikely to occur, leading to a similar decline in fear. The ability to model a complex and time-intensive clinical treatment through a simple laboratory paradigm enables researchers to examine different mechanisms of change and manipulations that could influence fear responses with greater specificity and efficiency than is possible in a clinical setting.

In addition, the brain circuitry involved in fear conditioning and extinction is relatively well-understood and conserved across species. This enables research with potential clinical relevance to occur at numerous different levels, ranging from brain imaging studies with clinically anxious patients [5] to studies testing the effects of pharmacological blockade of particular neurotransmitters in animals [6], all while using the same basic experimental paradigm. Furthermore, clinically anxious patients show aberrant fear and safety learning processes in conditioning and extinction studies [7,8], enabling researchers to better model and target the learning and neural mechanisms that may be responsible for fear-based psychological disorders.

Despite the promise of such translational research on extinction, there are limitations to heavily relying on a bottom-up approach toward improving clinical treatments, many of which can be easily missed by basic researchers with little to no contact with clinical practice. In particular, the conscious experience of fear and the ability for humans to engage higher-order cognitive processes not easily modeled in animals (e.g., cognitive reappraisal, imagery, etc.) both complicates clinical treatment and offers additional opportunities for intervention. Accordingly, broadening the theoretical framework and methods used to study fear learning in laboratory settings could help to enhance the relevance of translational research to the treatment of fear-based psychological disorders.

We begin with a selective review of several key findings resulting from translational research on extinction in order to highlight the promise that such work has for better understanding fear-based disorders and improving clinical treatments. We then describe a number of the limitations of the extinction model, and provide an overview of alternative psychological processes that are important to consider for building improved models of anxiety treatment. We conclude with suggestions for how to bridge the gap between research in the laboratory and the clinic.

## 2. Successes of the Extinction Model

Research on fear extinction has undoubtedly led to a number of valuable insights into processes through which pathological fears and defensive responses can be reduced. These successes include an increasingly sophisticated understanding of the neural and learning mechanisms involved in fear extinction and anxiety treatment, as well as a number of treatment augmentation strategies that show promise for improving clinical interventions. Below we provide a brief overview of some of the most important successes of the extinction model thus far.

### 2.1. The Brain Circuitry of Fear Reduction

One significant accomplishment of translational research on fear and anxiety over the last several decades is the compilation of a detailed account of the brain circuitry involved in fear conditioning and extinction [9,10]. At the heart of this circuitry is the amygdala, which comprises a number of sub-nuclei that are key for the detection and response to threats [11]. Specifically, sensory information related to the CS and US is integrated within the lateral nucleus of the amygdala, which is then relayed to the central nucleus [12,13]. From the central nucleus, projections to the hypothalamus and brainstem initiate a host of automatic behavioral (e.g., freezing) and autonomic (e.g., increase in blood pressure) defensive responses [14]. Synaptic plasticity in the lateral nucleus of the amygdala following CS–US pairings enables such downstream effects to be prompted by just the CS, leading to the development of conditioned defensive responses [12]. Conditioned responses in humans also involve a network of anatomically distributed brain regions that include the dorsal anterior cingulate cortex, anterior insula and dorsal medial prefrontal cortex [15,16], which play a role in anticipatory threat responses and the subjective experience of fear and anxiety [17,18].

During extinction training, many of the brain regions implicated in fear conditioning are similarly activated [19], with additional involvement of areas of the prefrontal cortex implicated in the cognitive regulation of emotions [20], as well as the anterior cerebellum [19,21]. In particular, the inhibition of defensive responses after extinction training is thought to be driven by activity in the ventromedial prefrontal cortex (vmPFC) [10,22]. Lesion and single-cell recording studies have found the vmPFC to impact recall of fear extinction without impacting acquisition or within-session extinction [23,24], suggesting that its role is specific to the retrieval of extinction memory. This has been further corroborated by research showing that attenuated conditioned responding during extinction recall is correlated with greater concurrent vmPFC activity in functional imaging studies [25,26], as well as increased thickness of the vmPFC [27,28]. Also involved in extinction recall is the hippocampus, which is thought to modulate the expression of an extinction memory based on contextual information [29]. Specifically, hippocampal activity facilitates retrieval of an extinction memory in the extinction context through connections with the vmPFC and amygdala, but in the conditioning context may play a role in the renewal of fear [25,30,31].

Recent research has also begun to elucidate the role of synchrony in neuronal oscillations, or variations in the frequency of electrical signals, across brain regions involved in fear conditioning and extinction [32]. Synchrony in theta and gamma oscillations between the medial PFC, hippocampus and amygdala appears to be involved in the formation and recall of fear [32,33]. The learning of safety during extinction on the other hand, has been shown to be related to decreased theta synchrony in these regions, as well as differential theta and gamma oscillations infralimbic and prelimbic regions [34,35,36].

The identification of the neural circuits and corresponding oscillatory activity involved in fear and extinction learning has provided another avenue for investigating potential mechanisms of clinical fear and anxiety, as well as related treatment targets. For example, individuals with posttraumatic stress disorder (PTSD) have shown decreased activity in hippocampal-vmPFC networks during extinction recall [5] and renewal [37], and dysregulation within this circuit may underlie the impaired ability to use contextual information to modulate fear seen in PTSD [38]. Similar dysfunction in vmPFC-amygdala networks has been demonstrated across other fear and anxiety-based disorders [39], and identifying individual differences in such networks has some potential for predicting treatment response and personalizing medicine [40,41,42].

Furthermore, there is a growing body of research on neuromodulatory interventions such as transcranial magnetic stimulation and transcranial direct current stimulation, which can directly target neural activity in the brain regions implicated in extinction [43]. For instance, modulating neural activity in the medial PFC during or immediately after extinction has been shown to enhance extinction retention [44,45] and improve exposure outcomes among individuals with PTSD [46]. Another promising method of neuromodulation is a technique called decoded neural reinforcement, which involves rewarding unconscious neural representations of feared stimuli as a way of counter-conditioning fear [47]. Early studies have shown such an approach can decrease fear responses to laboratory conditioned fears [48] as well as pre-existing fears of specific animals (e.g., snakes, cockroaches) [49]. Although investigation of these techniques for fear and anxiety-based disorders is still in its early stages, such approaches show intriguing potential for augmenting extinction learning processes in the context of clinical treatments [50].

### 2.2. Mechanisms of Exposure and Return of Fear

An important insight that has developed out of the research on extinction circuitry is that fear memories are not erased through extinction, but rather their expression is inhibited by cortical processes [51]. Extinction involves learning another meaning of the CS (e.g., of safety), in addition to that learned during fear acquisition. Thus, the CS becomes ambiguous, with multiple possible meanings/consequences, and context gives clues as to whether the consequence will be aversive or otherwise [52]. In concert with behavioral research on animals, the above-mentioned studies on the inhibitory circuitry involved in extinction learning have helped to clarify the processes underlying the relative fragility of safety learning seen clinically [53,54]. Specifically, three instances of the return of conditioned responses demonstrate how feared associations are inhibited rather than eliminated entirely.

For one, *renewal* occurs when a CS is encountered in a context other than where extinction training took place, which can either be the original conditioning context (ABA renewal) or a novel context (ABC renewal). The return of fear seen in both scenarios suggests that retrieval of the inhibitory CS-no US memory formed during extinction is relatively context specific, as even in a novel context that is ambiguous with regard to safety, the memory of the CS–US pairing tends to prevail [55]. Importantly, the range of possible contextual changes leading to renewal is wide, from physical location of training [56] to virtual reality environments [57] and internal states such as being caffeinated [58]. Our understanding of renewal helps to explain why even after successful exposures in the therapy office that reduce a patient’s fear of, say, public speaking, fear responses can return when the patient faces a similar situation in a different context such as at work or school.

A second instance of return of fear is *spontaneous recovery*, which refers to the return of a conditioned fear response after time has passed since extinction training [59]. The passage of time can give way to a new “context,” and thus, it can be thought of as a type of renewal [60]. Continuing with the example of social anxiety, a patient who successfully gives a speech in front of a group with minimal levels of fear might find her fear returning in the same situation months later, particularly if she has not been in a similar situation for some time.

Lastly, *reinstatement* refers to when a subject encounters an unsignaled US (e.g., a shock), producing a fear response that reminds them of the conditioned response they felt when they encountered the CS, consequently leading to a return of fear to that CS [61]. For instance, if a socially anxious student who has overcome their fear of public speaking gets laughed at or judged by her classmates, reinstatement can occur. In such a scenario, the experience of the US (being judged) causes the patient’s anxiety of public speaking to return, likely because it serves as a reminder of the potential costs of public speaking.

Our understanding of these return of fear processes has informed strategies to enhance the retention of extinction learning, with potential implications for improving the efficiency of treatment and decreasing relapse. Specifically, the inhibitory learning framework suggests that return of fear can be reduced by (1) strengthening the formation of the extinction memory and (2) enhancing the likelihood of retrieving the safety memory after extinction has occurred [54]. Building off this framework, a number of strategies to reduce the return of fear have been examined in both laboratory and clinical contexts. For example, variability in the contexts and stimuli used during extinction training, which can produce a less context or stimulus-specific extinction memory, has been shown to attenuate renewal following laboratory extinction training [62,63,64] and exposure therapy with specific phobias [65,66,67]. Further, the use of a retrieval cue, which is a neutral stimulus present during extinction that can later enhance retrieval of the extinction memory, has shown promise in reducing return of fear in both laboratory [68,69,70] and clinical studies [71,72].

### 2.3. Pharmacological Augmentation of Treatment

One of the most fruitful areas of translational research in fear-based disorders has been the use of pharmacological interventions designed to enhance extinction learning. The most successful of these have targeted the N-methyl-d-aspartate (NMDA) receptors in the amygdala, which have been implicated in fear acquisition and extinction [73,74,75]. In rodents, NMDA receptor antagonist administration has been found to weaken within-session extinction learning when given prior to extinction training [73,76] and impair extinction retention when given immediately after extinction training [77]. Based on such results, researchers began investigating whether the NMDA agonist d-cycloserine (DCS) might enhance the effects of extinction when infused directly into the basolateral nucleus of the amygdala or administered systemically, with consistently successful results in pre-clinical animal studies [6,78,79].

The effects of DCS have also been extended to humans in the context of exposure therapy, with randomized, placebo-controlled trials demonstrating that pre-exposure administration of the drug can improve clinical outcomes in height phobia [80], social anxiety disorder [81] and panic disorder [82]. Although effects have been somewhat variable across studies, a recent participant-level meta-analysis [83] found that DCS is associated with a small augmentation effect on exposure therapy in clinical populations with anxiety, obsessive-compulsive, or posttraumatic stress disorders. Notably, this effect has been shown to be dependent on the success of extinction learning in both animal [78] and human studies [84]. Patients who receive DCS and report low fear at the end of an exposure session appear to show greater clinical improvement compared to placebo, while those who receive DCS but report higher fear at the end of an exposure session show less improvement compared to placebo [85,86]. Investigation of whether tailoring DCS administration based on end fear levels would provide maximal benefit is currently under way [87].

### 2.4. Targeting Reconsolidation of Fear Memories

Another area of burgeoning translational research is the modification of fear memories during memory consolidation and reconsolidation [88]. When a memory is initially formed or is retrieved, there is a brief window in which it can be disrupted, either pharmacologically [89] or behaviorally [90]. To take advantage of this, researchers have investigated the effects of cueing the memory of conditioning with a single presentation of the CS, and then conducting extinction within the reconsolidation window while the memory is still labile, a technique referred to as post-retrieval extinction [91,92]. The advantage of such a technique is that rather than inhibiting the original CS–US association, the memory of the association can be directly modified, potentially eliminating the risk of return of fear present with standard extinction. Variable results from studies comparing post-retrieval and standard extinction procedures show that harnessing reconsolidation processes is not without complications, but a meta-analytic research by Kredlow and colleagues [91] did show a small-to-moderate effect size (*g* = 0.40) in favor of the technique reducing return of fear in human studies. Translation of either pharmacological or behavioral manipulations of reconsolidation into clinical treatments has also shown mixed results (see [92] for a review), but several examples of positive findings [93,94] demonstrate promise for the clinical utility of such techniques.

## 3. Limitations of the Extinction Model

Any laboratory model of a clinical phenomenon will inevitably have limitations, and extinction is no exception [95]. Understanding these limitations is important in order to have realistic expectations about what translational research on extinction learning can offer anxiety treatment, and also to highlight opportunities for extinction research to become more relevant for improving clinical interventions. In the sections below, we describe a number of ways in which conditioning and extinction paradigms fall short of capturing the complexity of the development and treatment of fear-based psychological disorders. We also highlight strategies for potentially overcoming such limitations.

### 3.1. Development of Clinical vs. Laboratory Fears

One important disconnect between fear conditioning extinction paradigms and the development and treatment of anxiety disorders is that frequently there is no easily identifiable conditioning event that leads to the onset of clinical anxiety [96,97,98]. Even in specific phobias, where an aversive experience with the phobic stimulus would seem most obvious and memorable, research has found that only between 18.0% and 57.5% of individuals are able to recall a direct conditioning event as the cause of their phobia [99]. Although the absence of such a memory does not mean that a conditioning experience did not occur, the prevalence of such experiences suggests that for many individuals with anxiety disorders, their condition developed in a manner other than the direct conditioning modeled in laboratory conditioning paradigms. Other possibilities include vicarious conditioning [100], secondary conditioning [101], mental representation and rehearsal of CS–US associations [102], and fear acquisition through socially transmitted information [103].

Conceivably, standard conditioning and extinction procedures may serve as a better model for conditions such as posttraumatic stress disorder or specific phobias where a clear conditioning event has occurred. This has not been investigated empirically, however, and even in disorders with identifiable direct conditioning events, fear typically becomes generalized across a wide variety of related situations and stimuli [104,105]. Fortunately, experimental research has illuminated a number of different ways in which conditioned fears can be elicited in laboratory settings besides straightforward CS–US pairings, some of which may serve to better model the diverse ways in which clinical fears develop. For instance, conditioned fear can be elicited through verbal instructions about the likelihood of a CS being followed by an aversive US in the absence of any actual CS–US pairings (see [106] for a review). Conditioned responses to a neutral stimulus can also be elicited through a learned association with a separate stimulus that has been or later will be paired with an aversive US, (i.e., higher-order conditioning) [101,107]. Importantly, fearful and defensive responses elicited through instructed conditioning, second-order conditioning and other generalization procedures involve additional learning processes and corresponding neural mechanisms beyond those involved in directly conditioned fears [16,105,107]. Because these processes are likely to be in play in clinical disorders, extinction researchers ought to utilize the full array of paradigms available for developing conditioned fears if it wants to maximize the potential for findings to be translatable to clinical settings.

### 3.2. Fear-Relevance of Conditioned Stimuli

Another notable limitation of fear extinction research for translational purposes is the fact that the large majority of studies use arbitrary stimuli like shapes or inanimate objects (e.g., lamps) as conditioned stimuli [7,19]. Anxiety and fear-based disorders typically involve exaggerated fear responses to stimuli or situations that have a threat-relevant component to them. For example, the social situations feared among those with social anxiety disorder inherently involve some risk (via negative judgment of others) in a way that the neutral stimuli frequently used in conditioning paradigms do not. In experimental settings, individuals with anxiety disorders also consistently demonstrate covariation bias (i.e., overestimation of CS-UCS contingencies) when CS’s are related to their clinical fears [108]. Mimicking such a bias through the use of fear-relevant CS’s, even with non-clinical samples, may therefore be helpful in making extinction paradigms more relevant for the treatment of anxious patients.

Importantly, conditioning with fear-relevant stimuli (e.g., pictures of snakes and spiders, angry faces, etc.) appears to lead to more rapid fear acquisition [109] and more resistance to extinction than fear-irrelevant stimuli [110,111], even among healthy participants. Because of this, manipulations that appear to enhance extinction with more neutral conditioned stimuli may not necessarily generalize across paradigms. For instance, verbally instructing participants about the likelihood of a US has been shown to diminish conditioned threat responses to fear-irrelevant conditioned stimuli, but had no effect on fear-relevant stimuli [112]. Similarly, a meta-analysis on post-retrieval extinction effects found significantly reduced return of fear for studies using fear-irrelevant, but not fear-relevant, stimuli [91]. Accordingly, experimental designs using fear-irrelevant stimuli may not be as meaningful for understanding the extinction learning processes occurring in the context of clinical levels of anxiety.

When selecting possible conditioned stimuli, it is important to be aware that the increased resistance to extinction seen with fear-relevant stimuli may vary depending on their evolutionary relevance [110,113], with the strongest effects seen in studies using images of spiders and snakes [114]. Maximal biological preparedness (see [113]) may not always be a desirable aspect of a conditioned stimuli, however, given that most anxiety disorders do not have the same evolutionary origins as spider and snake phobias. Furthermore, images of spiders and snakes elicit not just fear but also disgust, which has been shown to be slower to habituate than fear and is more relevant to some disorders (PTSD, obsessive-compulsive disorder (OCD), certain specific phobias) than others [115]. Socially and culturally fear-relevant images such as angry or out-group faces or pictures of guns offer an alternative set of stimuli that avoid some of these complicating factors. Fear responses to such stimuli appear be more susceptible than spider and snake images to influence by higher-order cognitive processes [114], which are an important component of interventions with clinical fears and are desirable to model.

It should be noted that even when fear-relevant stimuli are not used, there may be advantages to using more complex and life-like stimuli that have characteristics of something one might develop fear to (e.g., an animal-like stimulus; [116]). In addition, the use of virtual reality paradigms is a promising direction for creating more realistic models of clinical fears and their treatment [117,118].

### 3.3. Meaning of the CS–US Association

Another consideration regarding the fear-relevance of conditioned stimuli is whether any conceptual connection exists between the CS and US. In clinical fears, pre-existing information about the relatedness of the CS and US may strengthen conditioned responses beyond whatever linkage is formed as a result of their temporal association. For example, an individual with PTSD from a car accident experiences fear when seeing cars in a busy intersection (the CS) not just because that is what he or she saw right before getting hit and seriously injured (the US), but also because those cars could be responsible for that same harm again. The CS (as well as the conditioned fear response) takes on a *meaning* of danger [119].

Experimental investigations comparing conditioning with conceptually related (angry face and scream) and unrelated (landscape and scream) CS–US pairings have found that related pairings show faster fear acquisition and greater resistance to extinction [120]. This suggests that conditioning paradigms with unrelated stimuli may not be adequately modeling the strength of conditioned associations that occur clinically. Several recent studies have attempted to go beyond a pre-existing conceptual link between the CS and US, instructing participants to imagine a scene or view a short film that provides additional meaning and complexity to the association [121,122]. For example, in a study by Kunze and colleagues [122] one group of participants viewed a short movie that included a scene of a woman screaming as she was forced to eat a piece of cake with nails in it, while the other group watched a neutral video. This was followed by conditioning procedures using an image of nails as the CS and the same woman screaming as the US. Participants who viewed the aversive video showed enhanced acquisition, delayed extinction, and stronger reinstatement effects compared to the neutral video group. Together with findings showing that disorder-specific CS–US combinations (e.g., faces followed by verbal insults in social anxiety disorder) lead to more durable conditioned fears [123,124], these results suggest that the presence of a meaningful conceptual relationship between temporally associated stimuli increases the difficulty of breaking such associations. Given the complexity of extinction of clinical fears, this may be desirable to model in many cases, but extinction studies incorporating such procedural elements are relatively rare.

### 3.4. Complexity of Generalization

Extinction paradigms typically use only a single reinforced CS (or occasionally two) during acquisition, extinction and tests of extinction retention. Anxiety-related disorders, however, involve fear responses to a wide array of situations and stimuli, and successful treatment requires generalization of safety learning across diverse situations. To model this, researchers can conduct extinction or return of fear tests with generalization stimuli (GS), which are stimuli that are sufficiently similar to the original CS to also produce a conditioned fear response [125,126]. Typically, GS consist of perceptual variations of the US, such as slight alterations in the color, size or shape of geometric shapes, and different gradations of dissimilarity can be tested to examine how far generalization occurs [127]. Perceptual similarity, however, is likely only a small part of what drives clinical fears (or their reduction during treatment) to generalize, and in some cases may not be relevant at all. The human capacity to understand the world through abstract categories, engage in conceptual reasoning, and draw connections through verbal or other symbolic relations means that generalization can occur in a wide array of ways beyond perceptual similarity [128,129]. For example, agoraphobia resulting from an aversive experience of being trapped in an elevator might generalize to fear of airplanes and crowds not due to perceptual commonalities, but because these situations are similarly perceived as difficult to escape from.

To better capture the complex ways in which generalization can occur, researchers have recently developed a number of different paradigms that model alternative forms of generalization. For example, Dunsmoor et al. [130] showed that pairing different stimuli from a category (e.g., tools, animals) with shock led to generalization of conditioned responses to other members of the category that had not previously been presented, and extinction of some members of a category can also generalize to unextinguished members [131,132]. Category-based generalization of conditioning and extinction has also been demonstrated in the context of a trained equivalence category, in which participants were taught that miscellaneous stimuli belonged to a particular group through a match-to-sample task [133]. Furthermore, conditioned responses to presentations of specific words have been shown to generalize to synonyms of such words [134], or even the same words in a different language [135], providing evidence of semantic generalization. Such paradigms offer promise for better modeling the way generalization of extinction may occur clinically than can be done with stimulus generalization.

Possible differences in the learning processes and brain mechanisms involved in such generalization may also lead to divergent results when testing manipulations designed to reduce the return of fear. For instance, an examination of post-retrieval extinction in a category generalization paradigm failed to prevent recovery of conditioned fear [131], whereas the same manipulation has shown benefits across paradigms involving a single CS+ [136,137,138]. The added complexity in more recently developed generalization paradigms is also in line with the recommendation by Lissek and colleagues [139] to place more emphasis on studying fear learning processes in “weak” or ambiguous situations, as opposed to “strong” situations where contingencies with aversive stimuli are much clearer. Because clinical anxiety involves exaggerated fear responses to situations where threat is possible but uncertain, enhancing uncertainty by examining extinction with a wide array of conceptually, semantically, and/or perceptually related stimuli [140], as well as lower reinforcement rates during conditioning, may prove valuable for providing better models of treatment of clinical fears.

### 3.5. Neglect of Avoidance Behavior

A final limitation of the extant research on extinction learning is the relative lack of emphasis on avoidance behavior as a fear-related outcome, a gap that has fortunately been gaining increased attention as of late [141,142,143,144,145]. Although avoidance is not always maladaptive, consistent avoidance of feared situations is a central contributor to the onset and maintenance of fear-based psychological disorders, and is also a primary treatment target [146,147]. In human fear conditioning research, however, most studies examine only physiological (e.g., skin conductance response, startle reflex) and/or subjective (e.g., expectancy, valence or fear ratings) outcomes in paradigms in which explicit avoidance responses are unavailable, thereby neglecting a crucial component of fear learning [142].

A number of recent studies have developed promising paradigms with which to study extinction of avoidance behavior [143,148,149,150]. These typically involve both standard Pavlovian fear conditioning procedures as well as instrumental avoidance learning, in which participants learn that a particular action prevents the occurrence of the US. The impact of extinction training, either with or without the availability of avoidance behaviors, can then be examined on the persistence of avoidance as well as subjective fear and physiological defensive responses. Avoidance behaviors are often associated with monetary or other costs in such paradigms, which provides an incentive to reduce such behaviors that mimics the cost avoidance has in clinical disorders. Also of relevance is evidence suggesting that subjective relief (a proxy for reward learning) drives avoidance behavior, and is worth measuring within avoidance paradigms [150]. Notably, LeDoux and colleagues [144] have recently argued that avoidance is driven not just by Pavlovian and instrumental learning processes, but also habit learning resulting from repeated use of avoidance behavior. Such behavior then becomes independent of the instrumental consequences of avoidance. Given the impact such habitual behavior can have on extinction learning and the treatment of clinical fears, habit-based avoidance appears to be an additional process that would be well-suited for investigation within experimental extinction paradigms.

### 3.6. A Note on Predictive Validity

Many of the suggestions in the sections above (and listed in Table 1 below) for improving the external validity of extinction paradigms have to do with making the experimental situation and associated fears more complex, realistic, and potentially more difficult to extinguish. The ultimate test for whether these sorts of experimental modifications are beneficial for translational purposes, however, is how well performance in a given paradigm predicts exposure treatment outcomes. This is relevant for evaluating not just the procedures and stimuli used across paradigms, but also the relative importance of different indicators of fear learning, as physiological, neurological, and subjective indices of performance may be differentially related to clinical outcomes.

Whether responses in conditioning and extinction paradigms predict clinical outcomes at all has just begun to be investigated, with only a handful of studies currently published [155,156,157,158,159]. Evidence of predictive validity was found in each of the studies, meaning at least one index of learning (conditioning or extinction) predicted a clinical outcome, though the measures and methods for calculating such indices varied greatly. This is certainly encouraging for the utility of the extinction model, particularly since these studies used relatively basic paradigms consisting of a single, fear-irrelevant CS+ that was not conceptually connected to the US (excepting [158]), and with samples whose anxiety (e.g., OCD, public speaking anxiety, various phobias) likely had diverse origins. Nonetheless, the strength of prediction was not always especially strong, and there is likely room for improvement in predicting clinical outcomes by more closely approximating certain elements of clinical fears and their treatment, starting with the suggestions listed in Table 1. This is a ripe area for further research, with intriguing possibilities for predicting and personalizing clinical treatment based on various indices of learning [160]. Furthermore, the selection of specific paradigms to predict clinical outcomes might be able to be personalized based on behavioral, neuropsychiatric, neuroanatomical, or other individual characteristics.

## 4. Broadening the Scope of Extinction

Beyond modeling the procedural elements and mechanistic processes involved in anxiety interventions as closely as possible, translational research on extinction also needs to keep in mind the types of changes and outcomes that are ultimately most important for those whom such research is designed to benefit. LeDoux and Pine [161] argue that much of basic science research on threat responding has focused on defensive circuits, behaviors and physiological responses, which they distinguish from the subjective experience of fear and anxiety that tends to bring people in for treatment. While it is often assumed that the defensive circuitry centered in the amygdala is responsible for subjective feelings of fear, well-replicated findings such as the de-synchrony between subjective and behavioral or physiological indices of fear [162,163,164] and the experience of fear among those with amygdala damage [165,166,167] suggest this is not the case. Rather, subjective experiences of emotion appear to arise from cognitive representations of higher-order states that result from integrating diverse sources of information throughout the brain and body [168]. Although the defensive circuitry emphasized in conditioning and extinction research certainly contributes to the conscious experience of fear, it does not determine it, and alternative cortical circuits likely play a role [169,170]. Accordingly, a sophisticated understanding of how to dampen hyperactivity of conserved defensive circuits is only a part of what is needed to improve the lives of those suffering from fear and anxiety.

One of the major implications of this two-system theory for translational research on extinction is that heavily focusing on defensive responses may limit the field’s potential for improving clinical treatments. Models of behavior, learning, and neurological processes based on animal work (e.g., [171,172]) have formed the foundation for much of the experimental research done on fear and anxiety, but such models are unable to adequately capture the array of higher-order cognitive processes (e.g., schema formation, reappraisal) that are unique to humans and play an instrumental role in the experience of emotion. As a result, translational research tends to focus on a rather narrow set of biological and psychological processes. At the same time, the idea that subjective experience of fear and anxiety constitute a distinct system suggests that targeting such higher-order cognitive processes within extinction research is an especially promising route for improving clinical interventions. Furthermore, clinicians and clinical researchers have spent decades developing and refining therapeutic techniques that are particularly attuned to the subjective experiences of anxious patients. Greater attention to these approaches from a basic science perspective could help improve both experimental and clinical research on extinction-related processes.

In the following sections and in Table 2, we provide an overview of a number of different constructs and processes that are that are integral parts of clinical treatments of fear and anxiety, but have received comparatively little attention in basic research on fear and defensive responses. In particular, we emphasize how these techniques have been or might be integrated into extinction paradigms to improve the formation, retention and generalization of safety learning, thereby reducing heightened reactivity to conditioned fears.

### 4.1. Cognitive Reappraisal

Reappraisal is based on the idea that humans are constantly making interpretations of ambiguous scenarios in the world, and that these interpretations influence how one feels. Individuals with clinical levels of fear or anxiety characteristically make appraisals that overestimate the likelihood of harm, for instance overestimating the danger of air travel after hearing the news of a plane crash [178]. By adopting an alternative perspective on a situation or considering additional information that may have initially been ignored (e.g., considering the vast number of planes landing safely), anxiety responses can be modulated [179]. Cognitive reappraisal is a cornerstone of many effective treatments for anxiety and fear-based disorders, for instance cognitive processing therapy for PTSD [180].

Although relatively few studies have been published, reappraisal has been shown to lead to reduced responses to conditioned fear in the absence of extinction training, even after a 24-hour delay [173,181,182]. In these studies, after repeated pairings of either spider or snake images with shock, participants were instructed to brainstorm alternative ways of thinking about the CS+ images, and were told that focusing exclusively on the shock can increase anxiety. Reappraisal can also be utilized during extinction training. In a study by Blechert et al. [123], neutral faces were followed by short videos of the same face insulting participants, and then during extinction participants were instructed to generate alternative explanations for the person’s prior rudeness when viewing the neutral face. Reappraisal led to reduced conditioned negative valence in healthy and socially anxious participants after extinction, and also eliminated the enhanced conditioned responding seen in socially anxious participants after regular extinction, such that they responded similarly to healthy controls. Thus, reappraisal appeared to help socially anxious participants overcome a deficit in extinction learning, which is particularly interesting in the context of research showing that reappraisal and extinction rely on overlapping brain circuitry [20]. Further research on the potential for adjunctive reappraisal processes to augment extinction learning in clinical populations is clearly warranted.

A relatively unexplored area in which reappraisal may be beneficial is in regard to extinction generalization. Techniques designed to enhance generalization such as retrieval cues [49], stimulus and context variability [183] or attentional manipulations [184] tend to rely on implicit training procedures in which an individual’s conscious awareness and active participation in the technique is not necessary. However, evidence shows that top-down reasoning can strongly influence generalization [185,186]. When confronted with a context or stimulus change after extinction, then, explicitly directing attention toward indicators of safety (e.g., commonalities with the extinction experience) and encouraging appraisals about the relevance of prior learning to the present stimuli may help to further foster generalization.

### 4.2. Distress Tolerance

Another potential mechanism underlying exposure therapy is distress tolerance, which is defined as the willingness to tolerate unpleasant physical or mental states [187]. Clinically anxious individuals frequently have beliefs about their inability to tolerate the distress involved in facing their fears, leading to chronic avoidance. Demonstrating that distressing feelings of fear and anxiety are tolerable and will not lead to irreparable harm, then, is a central goal of exposures [54,188], and improvements in fear tolerance have been shown to mediate treatment outcomes [189].

Within conditioning and extinction paradigms, distress tolerance may be relevant to evaluations of the negative valence of the CS, the conditioned response (CR), or the US. Viewing the fear and physiological arousal related to these stimuli as intolerable is likely to make them more resistant to extinction, whereas greater toleration could enhance it. Consistent with this idea, Davey [190] proposed that the mental representation of US, (i.e., how unpleasant or aversive one believes it to be) can influence the strength of the CS–US association, and a number of studies have shown that the revaluating the US to be less unpleasant can reduce fear renewal [123,174,191,192]. Similarly, changes in CS valence, though more resistant to extinction, have been shown to predict return of fear [193,194]. Occasional presentations of the US during extinction have also been shown to protect from return of fear, which has been suggested to result from ‘physiological toughness’ [195], though the mechanisms through which this occurs are unclear.

Distress tolerance may be particularly relevant for avoidance paradigms, as the willingness to experience unpleasant states is likely to strongly influence the decision to avoid potentially aversive stimuli. Consistent with this notion, a study by Vervliet and colleagues [150] found that low distress tolerance led to persistent avoidance and sustained feelings of relief after extinction, which were minimally influenced by subsequent omissions of the US. Evidence suggests that distress tolerance can be improved through mindfulness interventions that foster a present-focused and nonjudgmental attitude toward emotional experience [196]. Additionally, psychoeducation and interoceptive exposure designed to demonstrate the benign nature of anxious feelings [197] can be beneficial. Applying such techniques as a form of US, CS or CR revaluation could be a fruitful future direction for enhancing the strength and durability of extinction.

### 4.3. Self-Efficacy

Closely related to the concept of distress tolerance is self-efficacy, which refers to the belief in one’s ability to cope with situational demands and perform well in the face of challenge [198]. Within the context of exposure, self-efficacy beliefs about effectively coping with anxious sensations and phobic situations can enhance one’s willingness to approach feared situations [199,200], and improvements in coping self-efficacy have been shown to mediate symptom reduction over the course of treatment [201,202,203]. A study by Zlomuzica and colleagues [175] built upon these clinical findings by experimentally manipulating self-efficacy before extinction training. Specifically, one group of participants received feedback that their responses earlier in the study meant they possessed excellent coping ability. At the end of extinction training, this group was equally likely to expect the US (i.e., no difference in contingency ratings) as a control group, but showed superior reductions in physiological responding and conditioned negative valence of the CS+. Relevant follow-up questions regarding this important finding include whether such an effect can be replicated with a clinical population that might have lower baseline coping self-efficacy, whether a behavioral manipulation of self-efficacy might have a similar impact, and if self-efficacy beliefs can also impact the return of fear (e.g., after a context change or on the subsequent day).

These findings also have implications for the way in which placebo effects might be harnessed. Although researchers typically attempt to tightly control for expectancy effects on outcomes, enhancing expectations of improvement in clinical practice can lead to improved outcomes [204]. The attributions patients make for their improvement are important, however, as attributing anxiety reductions to an external cause (e.g., medication) can make one more vulnerable to relapse [205]. Accordingly, researchers should attend to participant expectations about extinction augmentation strategies (e.g., neuromodulation) and their interaction with coping self-efficacy, as increased self-efficacy is likely to lead to a more durable reduction of fear.

### 4.4. Mental Imagery

Another uniquely human ability that has relevance to fear extinction is the ability to form mental representations or images of real-life stimuli. Mental imagery can evoke powerful emotional responses by influencing the same emotional systems in the brain responsive to real-life perception, and also enabling contact with emotional memories [206]. As a result, imagery is frequently used clinically in order to conduct exposures to situations that would be unethical or impossible to create in real-life, such as traumatic memories in PTSD [207] or worst-case future scenarios in generalized anxiety disorder and OCD [208]. In addition, changing the image of an aversive memory into more benign form, a procedure called imagery rescripting, has proven to be an effective treatment for various disorders in which highly negative memories are prevalent [209].

Imagery can feasibly play a role in the context of conditioning and extinction through mental rehearsal of associative relationships, and through the modification of such relationships in imagination. Mentally rehearsing the relationship between an experienced CS–US pairing can strengthen conditioning [210], and conditioning can even occur with an imagined US or CS–US pairing [211]. Conversely, imagined extinction of a CS with no US after has been shown to reduce physiological responses to subsequent CS exposure, and also showed similar neural activity to extinction with real-life unreinforced CS presentations [176]. Another potentially fruitful use of imagery is to mentally rehearse the memory extinction training in order to increase the likelihood of its retrieval, a method that has shown some promise in clinical literature [212,213], but has not been tested in an extinction paradigm.

Lastly, two studies have shown that imagery rescripting can enhance extinction of conditioned fear [121,214]. In a study by Dibbets and colleagues [121], participants imagined a scene in which a child was hit by a car and died in their arms, and then went through a conditioning phase involving car image paired with an image of a mutilated child. Imagining a less aversive ending to the story (the child recovering) led to reduced renewal of US expectancy and decreased negative valence of the US. Although more research is needed to understand the exact mechanisms, imagery rescripting may represent an alternative method of harnessing memory reconsolidation processes [209]. Using imagery not just to enhance learning but to update fear memories is a ripe area for future investigation. In fact, a recent study found that imagining a CS prior to extinction training eliminated return of fear to that CS following reinstatement, providing support for the idea that mental imagery is sufficient to reactivate a fear memory and open the reconsolidation window, thereby enabling that memory to be modified [215].

### 4.5. Verbal Processing

The ability for humans to use language to symbolically represent emotional states and form narratives about past experiences can also impact fear and anxiety responses [216,217]. Although the impact of verbal processing on emotions has largely taken place in the context of research on autobiographical episodic memory, there is reason to believe that linguistic processes can have impact fear learning as well [218]. In fact, a brief therapy protocol for PTSD consisting of repeatedly writing about the details of a traumatic event, as well as the surrounding thoughts and emotions has shown equivalent results to much longer gold-standard PTSD treatments [219], highlighting the powerful effects that verbal processing can have on conditioned fear. Although it has yet to be explicitly examined in the context of an extinction paradigm, the effects of labeling one’s emotions, and in some instances one’s feared outcomes, on exposure outcomes has been investigated in a number of studies [177,220,221], with results suggesting that such a technique can improve reductions in fear and defensive responses. Moreover, the regulatory effects of affect labeling appear to work via the same inhibitory connections between the PFC and amygdala that are implicated in extinction learning [222]. The effect of affect labeling on emotions experienced during conditioning and extinction, then, appears to be a promising area for further exploration.

## 5. Summary and Conclusions

In this paper, we began by highlighting some of the primary successes of extinction learning as a translational model for the treatment of anxiety and fear-based disorders. These include the identification of brain regions critical for inhibiting conditioned defensive responses, the development of a framework and corresponding techniques for improving the retention of extinction learning, the translation of the medication DCS from the animal laboratory to the clinic, and a growing understanding of how to target reconsolidation processes in concert with extinction to edit fear memories. Each of these areas of success has opened up new lines of research that have the potential to advance treatment, many of which involve novel ways of manipulating the brain circuitry involved in extinction [50].

However, we have also argued that a number of important limitations regarding the way extinction research is conducted need to be addressed in order to make the most of its translational potential. This involves identifying ways in which paradigms can better encapsulate key elements of the development and treatment of clinical disorders, a process which is thankfully in progress. It also involves broadening the focus of research beyond the defensive circuits, behaviors and physiological responses that influence but do not determine subjective fear, as well as better understanding and utilizing the higher-order cognitive processes and constructs that are used and studied in applied clinical research. This means that future research on translational interventions like device-based neuromodulation should target not just defensive survival circuits and non-conscious processes, but also those brain regions involved in subjective experience and appraisal of fear. In addition, such approaches are most likely to make a difference in patients’ lives if they are used while also considering the higher-order cognitive processes and constructs discussed in this paper (e.g., reappraisal, self-efficacy, etc.), and drawing upon related techniques that can lead to clinical improvement. 

The treatment of clinical fear and anxiety is a complex process that is undoubtedly difficult to capture in a laboratory model. Basic scientists studying animals and humans have important tools to better understand and model such complexity, but the knowledge of clinicians and clinical researchers about the complexities of applying and tailoring treatments [223] are also essential. Ultimately, more bidirectional communication of ideas and frameworks between basic scientists studying extinction and clinical researchers researching anxiety treatment is the most promising way forward. Clinician-researchers can help basic scientists to understand what is most likely to be relevant and translatable clinically, and in exchange basic scientist-clinicians can model and test clinical hypotheses at the level of their basic learning and neural mechanisms. Such collaboration should enable researchers to better synthesize knowledge about how to influence and utilize both defensive survival circuits and higher-order cognitive processes to enhance extinction learning, and then translate such knowledge into improved clinical treatments.

## Figures and Tables

**Table 1 brainsci-09-00164-t001:** Strategies for enhancing the external validity of extinction paradigms. CS, conditioned stimulus; US, unconditioned stimulus; PTSD, posttraumatic stress disorder.

Strategy	Exemplar(s)	Open Questions
Test extinction after indirect forms of conditioning (e.g., second-order, vicarious, instructed)	[151,152]	Is extinction with directly conditioned fears a more valid model for disorders with clear conditioning events (e.g., PTSD)?
Use fear-relevant conditioned stimuli, tailored to clinical population if relevant	[153,154]	Do methods of enhancing extinction hold up with fear-relevant stimuli?
Use virtual reality paradigms	[117,118]	Are conditioned fears in virtual reality paradigms more resistant to extinction?
Use CS–US pairings that are conceptually related	[121,123]	How is the impact of conceptually related stimuli influenced by the use of a narrative connecting the CS and US?
Test extinction and return of fear with perceptually, conceptually, and/or semantically related generalization stimuli	[131]	Do methods of enhancing generalization work differently across different type of generalization stimuli?
Include the availability of CS and/or US avoidance in extinction paradigms	[148,149]	How does the availability of avoidance behaviors at different stages of learning of influence extinction outcomes?
Examine prediction of clinical outcomes based on extinction of conditioned fears	[155]	What indices of learning from conditioning, extinction and return of fear tests are most predictive of clinical outcomes?

**Table 2 brainsci-09-00164-t002:** Constructs and processes to apply to extinction paradigms.

Mechanism	Definition/Use	Exemplar(s)
Cognitive Reappraisal	Adopting an alternative perspective on a situation additional information in order to modulate emotional response. Can be applied to either the CS or US.	[123,173]
Distress Tolerance	Willingness to tolerate unpleasant physical or mental states. May impact evaluation of CS, CR and/or US, as well as corresponding avoidance.	[150,174]
Self-efficacy	Belief in one’s ability to cope with situational demands and perform well in the face of challenge.	[175]
Mental Imagery	Creating a mental image of a different relationship between the CS and US.	[121,176]
Verbal Processing	Using language to label emotions and create a coherent narrative may help to reduce emotional responding to threat	[177]

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
