# Peer review of "From Extinction Learning to Anxiety Treatment: Mind the Gap"

_brainsci, 2019, doi:10.3390/brainsci9070164_

Round 1
Reviewer 1 Report
Overall, this is an excellent review paper regarding fear learning with an emphasis on fear extinction, and its relationship to anxiety disorders. It gives an excellent survey of the literature and includes insightful discussions of both lower-level mechanisms and higher-order processes.
I have minor suggestions that I believe could help ensure the review has some of the most recent research related to the topics covered.
In discussing the brain systems I would encourage the authors to consider the findings of two of the meta-analysis they cite (Mechias et al. 2010; Fullana et al. 2018) as well as citing Fullana et al. (2015). All three meta-analysis emphasize non-amygdala, structures involved in fear conditioning and extinction. In particular, the regions that appear to show the larger effects are the anterior insula, which is not mentioned.
There are two recent articles by Gregoire and Greening, 2019, Cognition; 2019, Cognition and Emotion). The first demonstrates the potential role of imagery in fear reconsolidation. The second relates to the role of conceptual relationships in fear generalization by showing the fear conditioning generalizes across semantically matched but orthographically different words in two different languages in bilinguals.
Two recent papers by the lab of Hakwan Lau (Koizumi et al 2016; Taschereau-Dumouchel et al 2018) introduce novel approaches to fear extinction the rely on modern neuroimaging techniques (neural feedback) and aspects of mental imagery.
Author Response
Overall, this is an excellent review paper regarding fear learning with an emphasis on fear extinction, and its relationship to anxiety disorders. It gives an excellent survey of the literature and includes insightful discussions of both lower-level mechanisms and higher-order processes.
I have minor suggestions that I believe could help ensure the review has some of the most recent research related to the topics covered.
In discussing the brain systems I would encourage the authors to consider the findings of two of the meta-analysis they cite (Mechias et al. 2010; Fullana et al. 2018) as well as citing Fullana et al. (2015). All three meta-analysis emphasize non-amygdala, structures involved in fear conditioning and extinction. In particular, the regions that appear to show the larger effects are the anterior insula, which is not mentioned.
Response: We have elaborated on the results of these meta-analyses on lines 84-90:
“Conditioned responses in humans also involve a network of anatomically distributed brain regions that include the dorsal anterior cingulate cortex, anterior insula and dorsal medial prefrontal cortex [Mechias et al., 2010; Fullana et al., 2016], which plays a role in anticipatory threat responses and the subjective experience of fear and anxiety [Kalisch et al., 2014; Paulus & Stein, 2006].
During extinction training, many of the brain regions implicated in fear conditioning are similarly activated [Fullana et al., 2018], with additional involvement of areas of the prefrontal cortex implicated in the cognitive regulation of emotions [20], as well as the anterior cerebellum [Fullana et al., 2018; Utz et al., 2015].”
There are two recent articles by Gregoire and Greening, 2019, Cognition; 2019, Cognition and Emotion). The first demonstrates the potential role of imagery in fear reconsolidation. The second relates to the role of conceptual relationships in fear generalization by showing the fear conditioning generalizes across semantically matched but orthographically different words in two different languages in bilinguals.
We now describe these studies on lines 345-347:
“Furthermore, conditioned responses to presentations of specific words have been shown to generalize to synonyms of such words [135], or even the same words in a different language [Gregoire & Greening, 2019a], providing evidence of semantic generalization.”
And lines 567-570
“In fact, a recent study found that imagining a CS prior to extinction training eliminated return of fear to that CS following reinstatement, providing support for the idea that mental imagery is sufficient to reactivate a fear memory and open the reconsolidation window, thereby enabling that memory to be modified [Gregoire & Greening, 2019b].”
Two recent papers by the lab of Hakwan Lau (Koizumi et al 2016; Taschereau-Dumouchel et al 2018) introduce novel approaches to fear extinction the rely on modern neuroimaging techniques (neural feedback) and aspects of mental imagery.
Response: We now describe this line of research on lines 123-127
“Another promising method of neuromodulation is a technique called decoded neural reinforcement, which involves rewarding unconscious neural representations of feared stimuli as a way of counter-conditioning fear [Taschereau-Dumouchel et al., 2018a]. Early studies have shown such an approach can decrease fear responses to laboratory conditioned fears [Koizumi et al., 2016] as well as pre-existing fears of specific animals (e.g. snakes, cockroaches) [Taschereau-Dumouchel et al., 2018b].”
Reviewer 2 Report
The work described emphasizes important limitations of the extinction model that translational approaches need to take into account, specially with regards to critical differences between laboratory and clinical studies. For instance aspects such as the use of fear relevant stimuli and conceptually related CS-US in paradigms are emphasized.
Moreover, the present perspective includes a set of strategies/constructs that have the potential to improve fear extinction rates.
Comments:
-Key brain structures corresponding to the circuitry of fear reduction are highlighted (lines 74-105), nevertheless the authors fail to emphasize the important role of neural correlates in the processes that determined such structures.
-Due to the relevance of neural oscillatory correlates for extinction of fear and its prospective use in translational approaches (See for instance Trenado et al. 2018 Neural Oscillatory Correlates for Conditioning and Extinction of Fear), it should be useful to emphasize the use of neural correlates in combination with the observations and strategies provided in the present perspective so as to present a unified framework to overcome the limitations of fear extinction models in clinical settings.
-Lines 149-171, the pharmacological approach for augmentation of treatment is emphasized. It should be valuable to mention the introduction of novel approaches such as neuromodulation in combination with extinction paradigms.
-Line 224-227 stresses that extinction researchers ought to utilize the full array of paradigms available for developing conditioned fears in clinical settings. I would also argue that proper selection of specific paradigms should consider neuopsychiatric, functional and anatomical factors, by following the principles of personalized medicine in combination to behavioral and neuroscience recent findings.
-Lines 362-379 the authors bring into the attention of the reader a very important aspect, namely predictive validity. It would be valuable if the authors express their view as to how to improve prediction of clinical outcomes.
-Lines-481-496, the authors refer to the interesting aspect of self-efficacy. How does this aspect relates to the placebo, nocebo effect?
Author Response
The work described emphasizes important limitations of the extinction model that translational approaches need to take into account, specially with regards to critical differences between laboratory and clinical studies. For instance aspects such as the use of fear relevant stimuli and conceptually related CS-US in paradigms are emphasized.
Moreover, the present perspective includes a set of strategies/constructs that have the potential to improve fear extinction rates.
Comments:
-Key brain structures corresponding to the circuitry of fear reduction are highlighted (lines 74-105), nevertheless the authors fail to emphasize the important role of neural correlates in the processes that determined such structures.
Response: We address this in our response to the following two comments.
-Due to the relevance of neural oscillatory correlates for extinction of fear and its prospective use in translational approaches (See for instance Trenado et al. 2018 Neural Oscillatory Correlates for Conditioning and Extinction of Fear), it should be useful to emphasize the use of neural correlates in combination with the observations and strategies provided in the present perspective so as to present a unified framework to overcome the limitations of fear extinction models in clinical settings.
Response: We have included research on neural oscillatory correlates of extinction on lines 102-111:
“Recent research has also begun to elucidate the role of synchrony in neuronal oscillations, or variations in the frequency of electrical signals, across brain regions involved in fear conditioning and extinction [Trenado et al., 2018]. Synchrony in theta and gamma oscillations between the medial PFC, hippocampus and amygdala appear to be involved in the formation and recall of fear [Trenado et al., 2018; Likhtik, et al., 2014]. The learning of safety during extinction on the other hand, has been shown to be related to decreased theta synchrony in these regions, as well as differential theta and gamma oscillations infralimbic and prelimbic regions [Fitzgerald et al., 2014; Meuller et al., 2014; Sperl et al., 2018].
The identification of the neural circuits and corresponding oscillatory activity involved in fear and extinction learning has provided another avenue for investigating potential mechanisms of clinical fear and anxiety, as well as related treatment targets.”
-Lines 149-171, the pharmacological approach for augmentation of treatment is emphasized. It should be valuable to mention the introduction of novel approaches such as neuromodulation in combination with extinction paradigms.
Response: We have now added a paragraph of neuromodulatory approaches on lines 118-129:
“Furthermore, there is a growing body of research on neuromodulatory techniques such as transcranial magnetic stimulation and transcranial direct current stimulation, which can directly target neural activity in the brain regions implicated in extinction [Gouveia et al., 2019]. For instance, modulating neural activity in the medial PFC during or immediately after extinction has been shown to enhance extinction retention [van’t Hout et al., 2017; Raij et al., 2018] and improve exposure outcomes among individuals with PTSD [Isserles et al., 2013]. Another promising method of neuromodulation is a technique called decoded neural reinforcement, which involves rewarding unconscious neural representations of feared stimuli as a way of counter-conditioning fear [Taschereau-Dumouchel et al., 2018a]. Early studies have shown such an approach can decrease fear responses to laboratory conditioned fears [Koizumi et al., 2016] as well as pre-existing fears of specific animals (e.g. snakes, cockroaches) [Taschereau-Dumouchel et al., 2018b]. Although investigation of these techniques for fear and anxiety-based disorders is still in its early stages, such approaches show intriguing potential for augmenting extinction learning processes in the context of clinical treatments [Lebois et al., 2019].”
And also discuss how these approaches might be best utilized in our summary and conclusions section, lines 595-610:
“Each of these areas of success has opened up new lines of research that have the potential to advance treatment, many of which involve novel ways of manipulating the brain circuitry involved in extinction [Lebois et al., 2019]…
…This means that future research on translational interventions like device-based neuromodulation should target not just defensive survival circuits and non-conscious processes, but also those brain regions involved in subjective experience and appraisal of fear. In addition, such approaches are most likely to make a difference in patients’ lives if they are used while also considering the higher-order cognitive processes and constructs discussed in this paper (e.g. reappraisal, self-efficacy, etc.), and drawing upon related techniques that can lead to clinical improvement.”
-Line 224-227 stresses that extinction researchers ought to utilize the full array of paradigms available for developing conditioned fears in clinical settings. I would also argue that proper selection of specific paradigms should consider neuopsychiatric, functional and anatomical factors, by following the principles of personalized medicine in combination to behavioral and neuroscience recent findings.
Response: We have acknowledged this point on lines 410-412:
“Furthermore, the selection of specific paradigms to predict clinical outcomes might be able to be personalized based on behavioral, neuropsychiatric, neuroanatomical, or other individual characteristics.”
-Lines 362-379 the authors bring into the attention of the reader a very important aspect, namely predictive validity. It would be valuable if the authors express their view as to how to improve prediction of clinical outcomes.
Response: We describe our view on how to improve predictive validity on lines 390-397:
“Many of the suggestions in the sections above (and listed in Table 1) for improving the external validity of extinction paradigms have to do with making the experimental situation and associated fears more complex, realistic, and potentially more difficult to extinguish. The ultimate test for whether these sorts of experimental modifications are beneficial for translational purposes, however, is how well performance in a given paradigm predicts exposure treatment outcomes. This is relevant for evaluating not just the procedures and stimuli used in across paradigms, but also the relative importance of different indicators of fear learning, as physiological, neurological, and subjective indices of performance may be differentially related to clinical outcomes.”
And lines 406-409
“Nonetheless, the strength of prediction was not always especially strong, and there is likely room for improvement in predicting clinical outcomes by more closely approximating certain elements of clinical fears and their treatment, starting with the suggestions listed in Table 1.”
-Lines-481-496, the authors refer to the interesting aspect of self-efficacy. How does this aspect relates to the placebo, nocebo effect?
Response: We write on lines 531-538:
“These findings also have implications for the way in which placebo effects might be harnessed. Although researchers typically attempt to tightly control for expectancy effects on outcomes, enhancing expectations of improvement in clinical practice can lead to improved outcomes [Price & Anderson, 2012]. The attributions patients make for their improvement are important, however, as attributing anxiety reductions to an external cause (e.g. medication) can make one more vulnerable to relapse [Powers et al., 2008]. Accordingly, researchers should attend to participant expectations about extinction augmentation strategies (e.g. neurmodulation) and their interaction with self-efficacy, as increased self-efficacy is likely to lead to a more durable reduction of fear.”
Reviewer 3 Report
It is an interesting paper. The bridge between basic science and translation into clinical practice is indeed very important and unfortunately a challenge for the community.
I Would like the authors to add a passage on the role of the physician scientist in her role. The autors suggest more communication between basic and clinical researchers. I agree. However, there are individuals out there who are both and they have a key role in driving these fields forward.
For inspiration I recommend. Pedroarena-Leal: Toward a symptom based neurostimulation for Gilles de la tourette Syndrome. Frontiers in Psychiatry 2017.
Author Response
It is an interesting paper. The bridge between basic science and translation into clinical practice is indeed very important and unfortunately a challenge for the community.
I Would like the authors to add a passage on the role of the physician scientist in her role. The autors suggest more communication between basic and clinical researchers. I agree. However, there are individuals out there who are both and they have a key role in driving these fields forward.
For inspiration I recommend. Pedroarena-Leal: Toward a symptom based neurostimulation for Gilles de la tourette Syndrome. Frontiers in Psychiatry 2017.
Response: We highlight the role of the physician-scientist (or we phrase it, the clinician-researcher) in our concluding paragraph, lines 611-622:
“The treatment of clinical fear and anxiety is a complex process that is undoubtedly difficult to capture in a laboratory model. Basic scientists studying animals and humans have important tools to better understand and model such complexity, but the knowledge of clinicians and clinical researchers about the complexities of applying and tailoring treatments [e.g. Pedroarena-Leal & Ruge, 2018] are also essential. Ultimately more bidirectional communication of ideas and frameworks between basic scientists studying extinction and clinical researchers researching anxiety treatment is the most promising way forward. Clinician-researchers can help basic scientists to understand what is most likely to be relevant and translatable clinically, and in exchange basic scientists clinicians can model and test clinical hypotheses at the level of their basic learning and neural mechanisms. Such collaboration should enable researchers to better synthesize knowledge about how to influence and utilize both defensive survival circuits and higher-order cognitive processes to enhance extinction learning, and then translate such knowledge in to improved clinical treatments.”
Round 2
Reviewer 3 Report
I feel a lot of important points have been added.